

# A total of 219 metagenome-assembled genomes of microorganisms from Icelandic marine waters

Clara Jégousse[1,2], Pauline Vannier[2], René Groben[2], Frank Oliver Glöckner[3,4] and Viggó Marteinsson[1,2]

[1] School of Health Sciences, University of Iceland, Reykjavik, Iceland
[2] Microbiology Group, Matís ohf., Reykjavik, Iceland
[3] Data at the Computing Center, Alfred Wegener Institute, Bremenhaven, Germany
[4] MARUM - Center for Marine Environmental Sciences,University of Bremen, Bremen, Germany

## ABSTRACT

Marine microorganisms contribute to the health of the global ocean by supporting the marine food web and regulating biogeochemical cycles. Assessing marine microbial diversity is a crucial step towards understanding the global ocean. The waters surrounding Iceland are a complex environment where relatively warm salty waters from the Atlantic cool down and sink down to the deep. Microbial studies in this area have focused on photosynthetic micro- and nanoplankton mainly using microscopy and chlorophyll measurements. However, the diversity and function of the bacterial and archaeal picoplankton remains unknown. Here, we used a co-assembly approach supported by a marine mock community to reconstruct metagenome-assembled genomes (MAGs) from 31 metagenomes from the sea surface and seafloor of four oceanographic sampling stations sampled between 2015 and 2018. The resulting 219 MAGs include 191 bacterial, 26 archaeal and two eukaryotic MAGs to bridge the gap in our current knowledge of the global marine microbiome.

## INTRODUCTION

Marine microorganisms are crucial to the global ecosystem as they regulate the carbon cycle (*Azam, 1998*; *Falkowski, Fenchel & Delong, 2008*) and support the marine food web (*Pomeroy, 1974*; *Azam et al., 1983*). The study of microorganisms within complex environments, such as the ocean, was accelerated by the emergence of sequencing technologies. In particular, metagenomics—the study of the total genetic material recovered from an environmental sample—have provided previously unavailable information on the functional diversity and ecology of the microbial communities within their environments (*Hugenholtz & Tyson, 2008*; *Quince et al., 2017*).

Large-scale metagenomics projects, such as the Global Ocean Sampling (*Venter et al., 2004*; *Rusch et al., 2007*), Ocean Sampling Day (*Kopf et al., 2015*) and Tara Oceans (*Sunagawa et al., 2015*; *Sunagawa et al., 2020*), have provided fascinating new insights, but also revealed the gaps in our knowledge of marine microbial species, their

Corresponding author
Viggó Marteinsson, viggo@matis.is

geographical distribution, and their organisation in complex and dynamic communities. These and other large-scale initiatives have so far not covered the oceanic regions around Iceland, a complex marine environment that is characterized by distinct water masses and powerful currents: the cold Polar Water of the East Greenland Current and the Arctic Water of the East Icelandic Current from the north and the warm North Atlantic Water of the Irminger Current from the south (*Malmberg, Valdimarsson & Mortensen, 1995*; *Valdimarsson & Malmberg, 1999*). Most microbial studies in Icelandic waters have so far been conducted with traditional methods, like chlorophyll measurements or microscopy, and were therefore mainly focused on larger heterotrophs and photosynthetic microorganisms (*Thórdardóttir, 1986*; *Gudmundsson, 1998*; *Astthorsson, Gislason & Jonsson, 2007*). To establish the baseline knowledge of microbial ecology in Icelandic marine waters, we assembled metagenomic sequence data into draft microbial genomes often called metagenome-assembled genomes (MAGs).

The recovery of MAGs opens the route to further analysis such as comparative genomics to understand the roles of these microorganisms within their community and ecosystem (*Sangwan, Xia & Gilbert, 2016*). MAGs are particularly valuable for yet uncultured marine lineages as they reveal the metabolic potential and environmental adaptation of these microorganisms and give clues about trophic interactions and ecology within the environment. Several marine metagenomic studies recovered MAGs from marine environments with—among others—136 MAGs from the Red Sea (*Haroon et al., 2016*), 290 from the Mediterranean Sea (*Tully et al., 2017*), and 2,631 from the global oceans with data harvested by Tara Oceans (*Tully, Graham & Heidelberg, 2018*).

Here, we report 219 MAGs from 31 samples collected in the Arctic Ocean north of Iceland and in the warmer Atlantic waters south of Iceland. The samples were collected between 2015 and 2018 at four established oceanographic sampling stations visited during six research cruises with two depths sampled at each station. A set of metadata is available for these samples following the best practices recommended by *Ten Hoopen et al. (2017)*, offering an opportunity to further understand the environmental conditions that shape the microbial communities in the waters off the Icelandic coasts.

## MATERIALS & METHODS

### Sampling

Seawater samples were collected between May 2015 and May 2018 from four stations, two in the North Atlantic Ocean, Selvogsbanki 2 and 5 (SB2 and SB5), and two in the Arctic Ocean, Siglunes 3 and 8 (SI3 and SI8) (Fig. 1A and Table 1). Sampling was conducted on board of the oceanographic research vessel Bjarni Sæmundsson RE 30 operated by the Icelandic Marine Research Institute (MRI) by collecting 5 L of seawater from the surface and the seafloor of the ocean, using Niskin bottles on a CTD rosette sampler. Seawater samples were directly filtered onto 0.22 µm Sterivex filter units (Merck Millipore) and immediately flash frozen in liquid nitrogen before stored at −80 °C until further processing (full workflow in Fig. 1B).
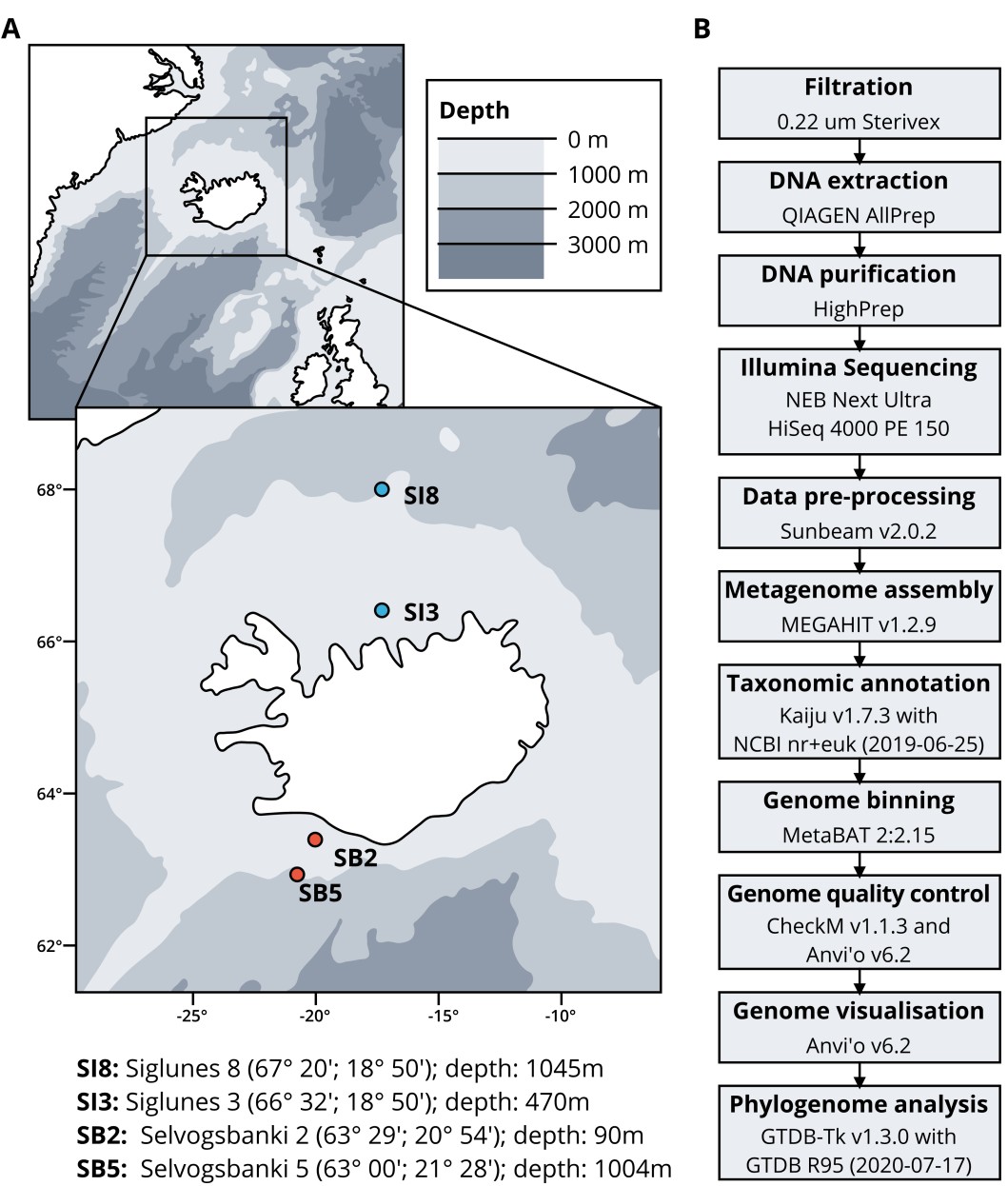

**SI8:** Siglunes 8 (67° 20'; 18° 50'); depth: 1045m
**SI3:** Siglunes 3 (66° 32'; 18° 50'); depth: 470m
**SB2:** Selvogsbanki 2 (63° 29'; 20° 54'); depth: 90m
**SB5:** Selvogsbanki 5 (63° 00'; 21° 28'); depth: 1004m

**Figure 1** (A) Sampling stations location and coordinates. (B) Workflow of bio-molecular processes and downstream analysis.

## Mock community

A marine mock community was included in the analysis for quality control, consisting of 20 bacterial and two archaeal species. Strains were cultivated according to Table 2. After 12 to 24 h of growth (to obtain 10e6 to 10e8 cell/ml), cells were counted on a Thoma cell BRAND (ref. 718020; 0.100 mm depth) to achieve a final concentration of 1.29 × 10e9 cell/L by dilutions. Synthetic seawater was prepared by adding 150 g of sea salts (Sigma-Aldrich, S9883 and 17.25 g of PIPES (Sigma-Aldrich, P1851) to 5 L of autoclaved MilliQ water.

**Table 1  Sampling dates and locations with corresponding seawater temperature and salinity.**

| Sampling date | Station ID | Latitude (dd.mm) | Longitude (dd.mm) | Depth (m) | Temperature (°C) | Salinity (PSU) |
|---|---|---|---|---|---|---|
| 23.05.2015 | SI8 | 67.9993 | −18.8313 | 1,045 | −0.481 | 34.913 |
| 30.05.2015 | SB5 | 62.9822 | −21.4737 | 0 | 7.632 | 35.195 |
| 30.05.2015 | SB5 | 62.9822 | −21.4737 | 1,004 | 4.391 | 34.998 |
| 23.05.2016 | SI8 | 68.0100 | −18.8247 | 0 | 1.632 | 34.869 |
| 23.05.2016 | SI8 | 68.0100 | −18.8247 | 1,045 | −0.431 | 34.914 |
| 31.05.2016 | SB5 | 62.9936 | −21.4839 | 0 | 8.147 | 35.113 |
| 31.05.2016 | SB5 | 62.9936 | −21.4839 | 1,004 | 4.722 | 35.017 |
| 21.05.2017 | SI8 | 68.0094 | −18.8325 | 1,045 | 2.700 | 34.852 |
| 21.05.2017 | SI8 | 68.0094 | −18.8325 | 0 | −0.381 | 34.914 |
| 22.05.2017 | SI3 | 66.5342 | −18.8378 | 470 | 5.517 | 34.492 |
| 22.05.2017 | SI3 | 66.5342 | −18.8378 | 0 | 0.151 | 34.906 |
| 30.05.2017 | SB5 | 62.9878 | −21.4800 | 1,004 | 8.477 | 34.761 |
| 30.05.2017 | SB5 | 62.9878 | −21.4800 | 0 | 4.801 | 35.009 |
| 09.08.2017 | SI3 | 66.5344 | −18.8419 | 0 | 9.980 | 34.310 |
| 09.08.2017 | SI3 | 66.5344 | −18.8419 | 470 | 0.190 | 34.900 |
| 09.08.2017 | SI8 | 68.0006 | −18.8375 | 1,045 | 7.640 | 34.650 |
| 09.08.2017 | SI8 | 68.0006 | −18.8375 | 0 | −0.370 | 34.910 |
| 18.08.2017 | SB2 | 63.4933 | −20.9569 | 0 | 12.000 | 33.700 |
| 18.08.2017 | SB2 | 63.4933 | −20.9569 | 90 | 8.470 | 34.940 |
| 18.08.2017 | SB5 | 62.9883 | −21.4867 | 0 | 12.200 | 34.980 |
| 18.08.2017 | SB5 | 62.9883 | −21.4867 | 1,004 | 4.730 | 35.010 |
| 16.02.2018 | SI3 | 66.5442 | −18.8400 | 470 | 0.044 | 34.901 |
| 16.02.2018 | SI8 | 68.0000 | −18.8386 | 0 | 0.533 | 34.640 |
| 16.02.2018 | SI8 | 68.0000 | −18.8386 | 1,045 | −0.410 | 34.914 |
| 18.05.2018 | SI8 | 68.0058 | −18.8256 | 0 | 1.355 | 34.727 |
| 18.05.2018 | SI8 | 68.0058 | −18.8256 | 1,045 | −0.428 | 34.914 |
| 20.05.2018 | SI3 | 66.5439 | −18.8406 | 0 | 5.108 | 34.894 |
| 29.05.2018 | SB2 | 63.4942 | −20.9008 | 0 | 7.625 | 34.913 |
| 29.05.2018 | SB2 | 63.4942 | −20.9008 | 90 | 7.298 | 35.031 |
| 29.05.2018 | SB5 | 62.9858 | −21.4731 | 0 | 7.740 | 35.042 |
| 29.05.2018 | SB5 | 62.9858 | −21.4731 | 1,004 | 4.488 | 34.978 |

The mock community was immediately treated in the same manner as the other seawater samples and filtered onto Sterivex filters for DNA extraction.

## DNA extractions

DNA was extracted from all samples using the QIAGEN AllPrep kit according to the manufacturer's instructions with modifications. Sterivex filters were aseptically removed from their plastic casing as described by *Cruaud et al. (2017)*. Filters were transferred to tubes containing 600 µl RTL buffer from the kit and 0.2 g of 0.1 mm zirconia/silica beads (BioSpec, cat. 11079101z) for mechanical disruption of the cells (bead-beating) using a Disrupt MixerMill MM400 by Retsch with the program P9 (300 Hz) three times for 10 s

Jégousse et al. (2021), PeerJ, DOI 10.7717/peerj.1112

Peerj

**Table 2  List of bacterial and archaeal species in the mock community.** Strains were obtained from the Icelandic Strain Collection and Records (ISCAR) or the German Collection of Microorganisms and Cell Cultures (DSMZ: https://www.dsmz.de/). Recipes for growth media can be found at if not otherwise indicated.

| Domain | Species name | % identity | Collection number | Growth parameters | Successfully reassembled |
|---|---|---|---|---|---|
| Bacteria | *Alteromonas naphthalenivorans* | 99.66% | ISCAR-05201 | Marine Broth, 22 °C, pH 6.8, aerobic condition | Yes |
| Bacteria | *Jeotgalibacillus marinus* | 100% | ISCAR-03118 | Marine Broth, 22 °C, pH 6.8, aerobic condition | No |
| Bacteria | *Geobacillus thermoleovorans* | 100% | ISCAR-00004 | 162 media, 65 °C, pH 7.0, aerobic condition | No |
| Bacteria | *Colwellia psychrerythraea* | 99% | ISCAR-05175 | Marine Broth, 22 °C, pH 6.8, aerobic condition | Yes |
| Bacteria | *Dietzia psychralcaliphila* | 99.52% | ISCAR-05191 | 92 media, 22 °C, pH 6.8, aerobic condition | No |
| Bacteria | *Escherichia coli* | 100% | ISCAR-02961 | LB media, 37 °C, pH 7.0, aerobic condition | Yes |
| Bacteria | *Pseudomonas salina* | 99.83% | ISCAR-05249 | Marine Broth media, 22 °C, pH 6.8, aerobic condition | No |
| Bacteria | *Marinobacter psychrophilus* | 99.84% | ISCAR-05186 | Marine Broth media, 22 °C, pH 6.8, aerobic condition | Yes |
| Bacteria | *Photobacterium indicum* | 100% | ISCAR-05002 | Marine Broth media, 22 °C, pH 6.8, aerobic condition | Yes |
| Bacteria | *Pseudoalteromonas neustonica* | 98.58% | ISCAR-05312 | 172 media, 22 °C, pH 6.8, aerobic condition | Yes |
| Bacteria | *Reinekea aestuarii* | 100% | DSM 29881 | Marine Broth media, 22 °C, pH 6.8, aerobic condition | No |
| Bacteria | *Reinekea marinisedimentorum* | 100% | DSM 15388 | Marine Broth media, 30 °C, pH 6.8, aerobic condition | Yes |
| Bacteria | *Rhodococcus kyotonensis* | 99.23% | ISCAR-05221 | Marine Broth media,22 °C, pH 6.8, aerobic condition | No |
| Bacteria | *Reinekea sp. 84* | 97.75% with *Reinekea marina* | ISCAR-05258 | Marine Broth media, 22 °C, pH 6.8, aerobic condition | No |
| Bacteria | *Sulfitobacter sp. 87* | 97.73% with *Sulfitobacter donghicola* | ISCAR-05261 | Marine Broth media, 22 °C, pH 6.8, aerobic condition | No |
| Bacteria | *Sulfitobacter donghicola* | 100% | DSM 23563 | Marine Broth media, 22 °C, pH 6.8, aerobic condition | Yes |
| Bacteria | *Sulfitobacter guttiformis* | 100% | DSM 11544 | Marine Broth media, 22 °C, pH 6.8, aerobic condition | Yes |
| Bacteria | *Sulfitobacter pontiacus* | 100% | DSM 10014 | Marine Broth media, 22 °C, pH 6.8, aerobic condition | Yes |
| Bacteria | *Sulfitobacter undariae* | 100% | DSM 102234 | Marine Broth media, 22 °C, pH 6.8, aerobic condition | No |
| Bacteria | *Thermus thermophilus* | 100% | ISCAR-03915 | 166 media, 65 °C, pH 7.0, aerobic condition | No |
| Bacteria | *Vibrio cyclitrophicus* | 100% | ISCAR-06209 | Marine Broth media, 22 °C, pH 6.8, aerobic condition | No |
| Archaea | *Pyrococcus abyssi* | 100% | DSM 25543 | YPS[1] media, 90 °C, pH 7, anaerobic condition, elemental sulfur | Yes |
| Archaea | *Thermococcus barophilus* | 100% | DSM 11836 | TRM[2], 85 °C, pH 6.5, anaerobic condition, elemental sulfur | Yes |

**Notes.**
Growth media recipes in: [1] *Erauso et al. (1993)* [2] *Marteinsson et al. (1999)*.

each, cooling down tubes in icy water in between each bead-beating step. DNA quality was assessed with a NanoDrop 1000 Spectrophotometer (ThermoFisher) and DNA was quantified with a Qubit fluorometer (Qubit DNA BR assay, Invitrogen).

## Library preparation and sequencing

High-throughput sequencing of the samples was performed by Genome Quebec using the HiSeq system (Illumina). Libraries were prepared using NEBNext UltraTM II DNA Library Prep Kit for Illumina (New England Biolabs) followed by sequencing on two lanes of an Illumina HiSeq 4000 PE150 system (Illumina) allocating 1/20 and 1/25 of a lane for each sample. Demultiplexing and conversion to FASTQ files were performed using bcl2fastq Conversion Software v1.8.4 (Illumina) resulting in 32 metagenomic datasets.

## Co-assembly and binning

The quality of the raw sequencing reads was assessed using FastQC v0.11.8 (*Andrews et al., 2012*) (Fig. S1). Quality control of the raw reads was performed with Sunbeam v2.0.2 (*Clarke et al., 2019*) which includes trimming with Trimmomatic v0.36 (*Bolger, Lohse & Usadel, 2014*), adapter removal with Cutadapt v2.6 (*Martin, 2011*) (parameters PE -phred33 ILLUMINACLIP: NexteraPE-PE.fa:2:30:10:8:true LEADING: 3 TRAILING: 3 SLIDINGWINDOW: 4:15 MINLEN: 36), removal of low complexity sequences using Sunbeam Komplexity (default parameter) and removal of contaminating human sequences using the Genome Reference Consortium Human Build 38 patch release 13 GRCh38.p13 (*Lander et al., 2001*; *Schneider et al., 2017*). Resulting quality-filtered metagenomic data were divided into surface and seafloor datasets as the surface of the ocean can be considered a different environment compared to the seafloor (Fig. S2). Both datasets also included the mock community. After quality filtering, MEGAHIT v1.2.9 (*Li et al., 2015*; *Li et al., 2016*) (parameters: –min-contig-len 1000 -m 0.85) co-assembled both datasets of samples with a minimum contig length of 1000 bp, resulting in two FASTA files of community contigs. Quality-filtered short reads from each sample were mapped back to the contigs of both co-assemblies respectively using Bowtie v2 (default parameters and –no-unal flag) (*Langmead & Salzberg, 2012*). The resulting SAM files were indexed and converted to BAM files with SAMTOOLS v0.3.3 (parameters: view -F 4 -bS) (*Li et al., 2009*). For both co-assemblies, the FASTA files containing the contigs were formatted with the script reformat-fasta from Anvi'o v6.2 (*Eren et al., 2015*). The two contigs databases (the surface and the seafloor databases) were generated with Anvi'o, BAM files were profiled and merged to the respective databases. Automated binning was performed using Anvi'o script anvi-cluster-contigs with default parameters with three binning algorithms: CONCOCT v1.1.0 (*Alneberg et al., 2013*), MaxBin2 v2.2.6 (*Wu, Simmons & Singer, 2016*), and MetaBAT 2 v2:2.15 (*Kang et al., 2019*). For all binning results, completeness and redundancy of the bins were estimated with Anvio's script anvi-estimate-genome-completeness which relies on CheckM v1.1.3 (*Parks et al., 2015*). Based on the comparison of the three binning algorithms, we selected the ''good quality bins'' from MetaBAT 2 with an estimated completion above 50% and an estimated redundancy below 10% according to standards suggested by *Bowers et al. (2017)*. The relative proportions of good quality bins in the total number of bins was assessed by $chi^2$ test.

**Table 3   Statistics summary of co-assemblies.**

|  | Surface | Seafloor |
|---|---|---|
| Total nucleotides | 1.06 Gb | 1.23 Gb |
| N50 | 2,382 bp | 2,327 bp |
| L50 | 83,272 bp | 114,549 bp |
| Number of contigs | 445,328 | 554,104 |
| Longest contig | 864,343 bp | 1,302,516 bp |
| Shortest contig | 1,000 bp | 1,000 bp |
| Number of contigs >10 kb | 8,521 | 8,306 |
| Number of genes (Prodigal) | 1,271,859 | 1,532,800 |

## Functional assignment, taxonomy and phylogenomic trees

We used PRODIGAL v2.6.3 (*Hyatt et al., 2010*) to identify Open Reading Frames (ORFs) within the contigs. The resulting ORFs were processed with Kaiju v1.7.3 (*Menzel, Ng & Krogh, 2016*) and NCBI nr+euk database (nr_euk 2019-06-25, 46GB, available for download at for taxonomic assignment. Beside the contig-based taxonomic assignment, we used GTDB-Tk v1.3.0 (Genome Taxonomy Database Toolkit) (*Chaumeil et al., 2019*) to construct two bacterial and two archaeal phylogenomic trees containing good quality MAGs (completeness ≥50%; contamination ≤10%) and Genome Taxonomy Data Bank (GTDB) R95 (released in July 2020) reference genomes to confirm taxonomic assignments of the MAGs (*Parks et al., 2018*). The trees were reconstructed using ARB (*Ludwig et al., 2004*) for comprehensive visualisation.

## Data availability

The raw Illumina sequencing paired-end reads are available in the ENA under project accession number PRJEB41565 (ERP125360). MAGs are available under accession numbers ERS5621908 to ERS5622126. Code is available at https://github.com/clarajegousse/.

## RESULTS

### Co-assemblies

The co-assembly of the 16 samples of the surface of the ocean yielded 445,328 contigs, with a minimal length of 1,000 bp, representing a total length of 1.06 Gb (1,060,942,783 nucleotides) with N50 of 2,627 bp and 1,271,859 gene calls (Table 3).

The co-assembly of the 17 samples of the seafloor of the ocean yielded 554,104 contigs, with a minimal length of 1,000 bp, representing a total of length of 1.23 Gb (1,233,390,295 nucleotides) with N50 of 2,327 bp and 1,532,800 gene calls (Table 3).

### Binning

A comparison of the three binning algorithms - CONCOCT, MaxBin2 and MetaBAT 2 - was conducted on the surface and seafloor co-assemblies based on the number of good quality bins (Fig. 2). Good quality bins have an estimated completion above 50% and an estimated redundancy (also called estimated contamination) below 10% (*Bowers et al., 2017*). The relative proportions of good quality bins is significantly different for the three
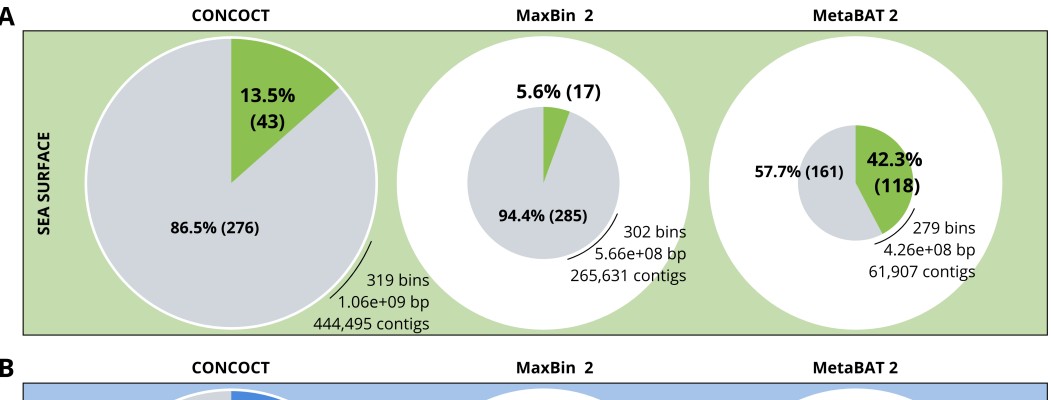

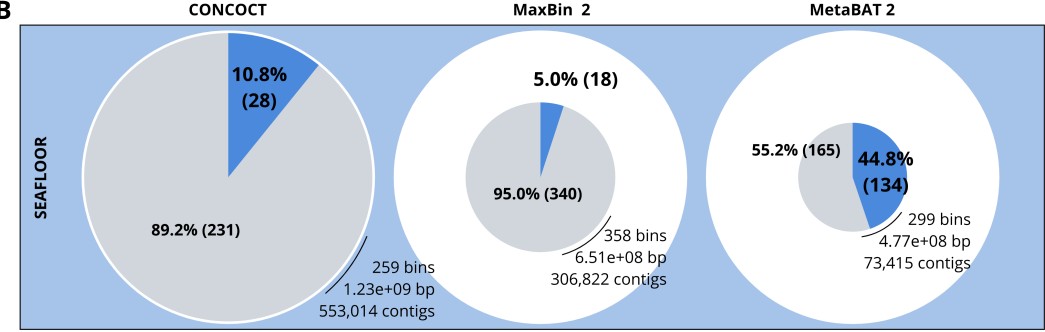

**Figure 2** **Binning comparison. Numbers of contigs binned and numbers of bad and good quality bins obtained with CONCOCT, MaxBin2 and MetaBAT 2 from the surface co-assembly (A) and the seafloor co-assembly (B).** Numbers of contigs binned is represented by the size of the pie plots. Numbers and percentages of bad quality bins and good quality bins are shown within the grey and coloured slices of the chart respectively. Good quality bins have an estimated completion above 50% and an estimated redundancy (also called estimated contamination) below 10% (*Bowers et al., 2017*).

binning methods ($\chi^2 = 135.23$, $df = 2$, $p$-value $< 2.2e{-}16$). The results of the binning showed that MetaBAT 2 resulted in a lower number of bins compared to CONCOCT and MaxBin2. Yet the number of good quality bins was much higher with MetaBAT 2 compared with CONCOCT and MaxBin2 (Table 4).

MetaBAT 2 gave the best results which were used for further analysis and shown in more detail in Fig. 3. Out of the 279 bins identified by MetaBAT 2 for the surface samples, 42.4% (118) of them are good quality bins that can be considered draft MAGs according to *Bowers et al. (2017)*. Within the 118 good quality MAGs (Fig. 3B), 16 represent genomes of organisms from the mock community and 102 are assembled from the surface seawater. In the same manner, out of the 299 bins identified by MetaBAT 2 for the seafloor samples, 45.81% (134) of can be considered good draft MAGs. Within the 134 good quality MAGs (Fig. 3D), 17 represent genomes of organisms from the mock community and 117 are assembled from the seawater at the seafloor. The relative proportions of MAGs out of the total number of bins is the same out of the two co-assemblies datasets ($\chi^2 = 0.27784$, $df = 1$, $p$-value $= 0.5981$) which means that the environments do not seem to impact significantly the number of MAGs. In the same manner, the relative proportions of MAGs associated to the mock community out of the total number of MAGs is the same in the two co-assemblies datasets ($\chi^2 = 0.0003$, $df = 1$, $p$-value $= 0.9858$).

**Table 4** Statistics summary of co-assemblies.

| Co-assembly | Binning method | Number of bins | Number of MAGs | Average completeness (%) | Average contamination (%) |
|---|---|---|---|---|---|
| Surface | CONCOCT | 319 | 43 | 45.15 | 49.23 |
| Surface | MaxBin2 | 302 | 17 | 25.77 | 13.30 |
| Surface | MetaBAT 2 | 279 | 118 | 44.12 | 3.46 |
| Seafloor | CONCOCT | 259 | 28 | 51.26 | 90.39 |
| Seafloor | MaxBin2 | 358 | 18 | 34.59 | 18.63 |
| Seafloor | MetaBAT 2 | 299 | 134 | 49.90 | 7.13 |

## Taxonomy

When excluding members of the mock community based on taxonomic assignment and differential coverage, we identified 102 MAGs reconstructed from the surface co-assembly and 117 MAGs from the seafloor co-assembly. The surface MAGs include two eukaryotes (*Bathycoccus* and *Micromonas*), 92 bacteria, and eight archaea while the seafloor MAGs include 99 bacteria, 18 archaea and no eukaryotes.

The surface co-assembly yielded a total of 92 bacterial MAGs (Fig. 4). These MAGs are members of seven phyla (number of MAGs in brackets): Proteobacteria (52), Bacteroidota (31), Actinobacteriota (2), Verrumicrobiota (2), Planctomycetota (2), SAR324 (1) and Cyanobacteria (1). The MAG within the Cyanobacteria phylum belongs to the genus *Synechococcus*. Within the phylum Actinobacteriota, we retrieved two MAGs: one from a member of the genus *Aquiluna* and one of the genus *Pontimonas*. We reconstructed two MAGs within the phylum Planctomycetota. The two MAGs within the Verrumicrobiota belong to the family Akkermansiaceae. The Bacteroidota phylum includes 31 MAGs reconstructed from the sea surface co-assembly. Most of these Bacteroidota MAGs belong to the Flavobacteriaceae family (18), including one representant of the genus *Polaribacter*. Many MAGs within the Flavobacteriaceae family are related to MAGs revealed by Tara Ocean Consortium such as Cryomorphaceae bacterium and Flavobacteriales bacterium (CFB group bacteria). We also reconstructed 52 MAGs belonging to the phylum of Proteobacteria, including nine Rhodobacteraceae, ten SAR86 and ten Porticoccaceae. Within the three MAGs of the Burkholderiales order, one is within the *Burkholderia* genus, and the two others belong to the Methylophilaceae family according to GTDB.

The seafloor co-assembly yielded a total of 99 bacterial MAGs spanning across 12 phyla: Proteobacteria (46), Verrumicrobiota (9), Bacteroidota (9), Marinisomatota (8), Actinobacteria (5), Planctomycetota (5), Gemmatimonadota (4), Nitrospinota (3), Chloroflexota (2), SAR324 (2), Myxococcota (1), Lactescibacterota (1). Six of these phyla include exclusively MAGs from the seafloor (Nitrospinota, Myxococcota, Gemmatimonadota, Marinisomatota, Chloroflexa, Lactescibacterota). Within the Proteobacteria, most of the MAGs belong to the Gammaproteobacteria class with 32 MAGSs while the remaining 14 are part of the Alphaproteobacteria. Five orders within the Proteobacteria exclusively include MAGs reconstructed from the seafloor co-assembly

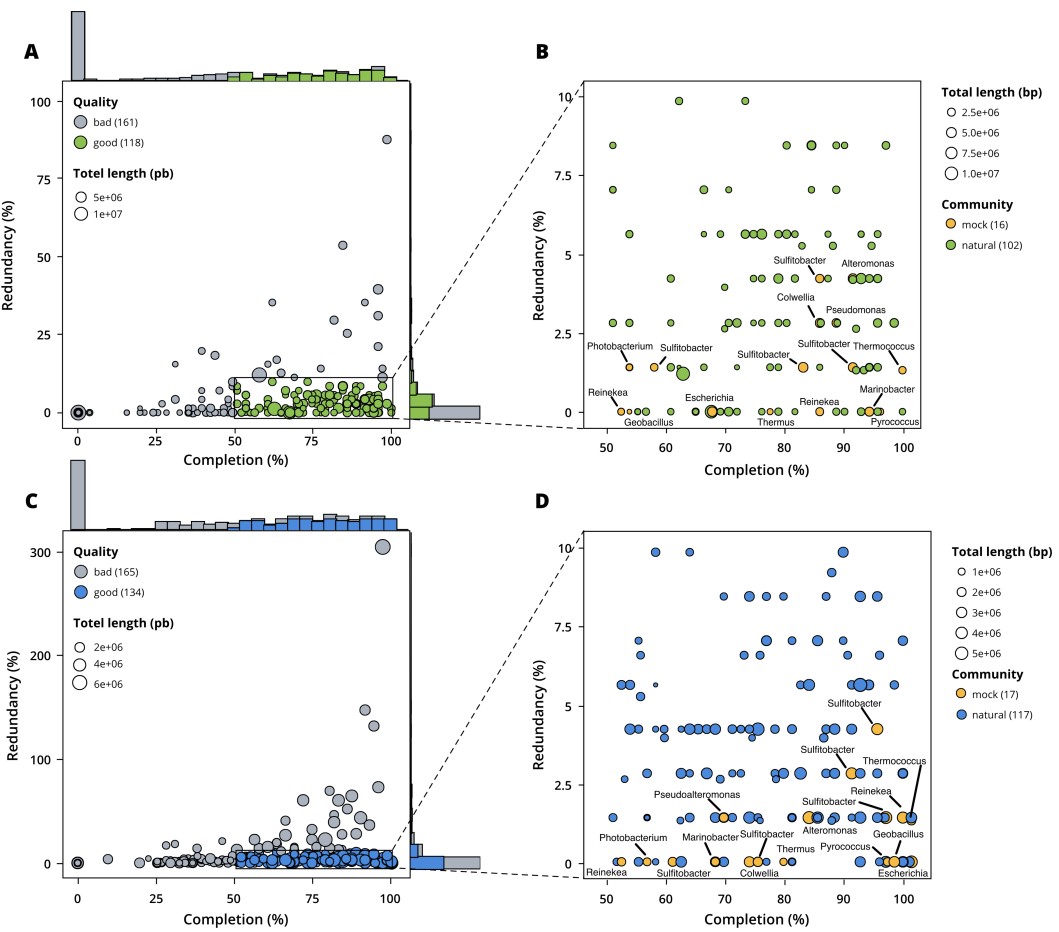

**Figure 3 Assessment of bin quality with the estimated completeness as a function of the redundancy.**
Bad quality bins (completeness below 50% and redundancy above 10%) are shown in grey while good
quality bins are in colours (green for surface, blue for seafloor samples). (A) A total of 279 bins obtained
with MetaBAT 2 from the surface co-assembly with 118 good quality bins. (B) Good quality bins from the
surface co-assembly with the identification bins corresponding to members of the mock community. (C)
A total of 299 bins obtained with MetaBAT 2 from the seafloor co-assembly with 134 good quality bins.
(D) Good quality bins from the seafloor with the identification of the bins corresponding to members of
the mock community.

(Rhizobiales, Rhodospirillales, TMED109, UBA10353, UBA4486) and none from the
surface co-assembly.

Out of the 21 bacterial species of the mock community, 12 of them were re-assembled
and given the correct taxonomic assignment down to species level (if available for the strain
used) for *Alteromonas sp.*, *Geobacillus marinus*, *Colwellia sp.*, *Escherichia coli*, *Marinobacter
sp.*, *Photobacterium sp.*, *Pseudoalteromonas sp.*, *Reinekea marinisedimentorum*, *Sulfitobacter
donghicola*, *Sulfitobacter guttiformis*, *Sulfitobacter pontiacus* and *Thermus thermophilus*.
However, some distinct species of the mock community that belong to the same genus do
not match any specific MAGs but seem to have been reassembled as one single MAG within
the genus in question, such as *Reinekea aestuarii* and *Reinekea sp. 84* as well as *Sulfitobacter*

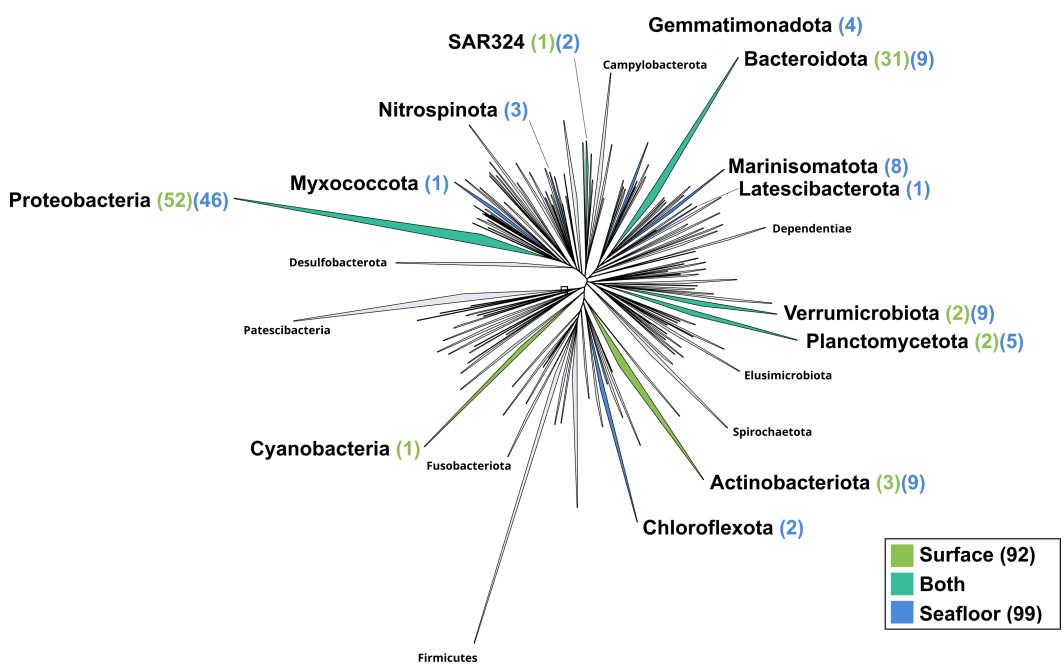

**Figure 4 Bacterial phylogenomic tree.** Distribution of the Marine Icelandic MAGs across 76 bacterial phyla from GTDB. The maximum likelihood tree was inferred from the concatenation of 120 proteins spanning a dereplicated set of 191,527 bacterial genomes (GTDB 05-RS95 released on the 17th July 2020) and the Marine Icelandic MAGs. Phyla containing MAGs from the surface seawater, seafloor or both are shown in green, blue or teal respectively. Number of Marine Icelandic MAGs from the surface and the seafloor in each phylum are indicated in between parenthesis in green and blue respectively.

*undariae* and *Sulfitobacter sp. 87*. The genomes of *Bacillus thermoleovorans*, *Dietzia sp.*, *Halomonas sp.* and *Vibrio cyclitrophicus* were not reassembled.

The surface co-assembly yielded only eight archaeal MAGs (Fig. 5), all within the Thermoplasmota phylum, including three MAGs within the genus MGIIb-O2 of the Thalassarchaeaceae family and five within the Poseidoniaceae family. The seafloor co-assembly resulted in 18 archaeal MAGs including one representant of the Thermoproteota phylum: this MAGs belongs to the UBA57 phylum within the order of the Nitrososphaerales. The 17 other archaeal MAGs are all comprised in the Thermoplasmatota phylum, within the class Poseidoniia, including representatives of the Poseidoniaceae and Thalassarchaeaceae families. The two archaeal members within the mock community (*Pyrococcus abyssi* and *Thermococcus barophilus*) were successfully reconstructed in both co-assemblies.

## DISCUSSION

Mock communities are used to quantify and characterise biases introduced in the sample processing pipeline (*Brooks et al., 2015*) and are indispensable to benchmark sequencing methods and downstream analysis (*Singer et al., 2016*; *Sevim et al., 2019*). Mock communities can also be used as a positive control for metagenomic studies. Our mock community confirmed that MetaBAT 2 was able to resolve genomes of species within

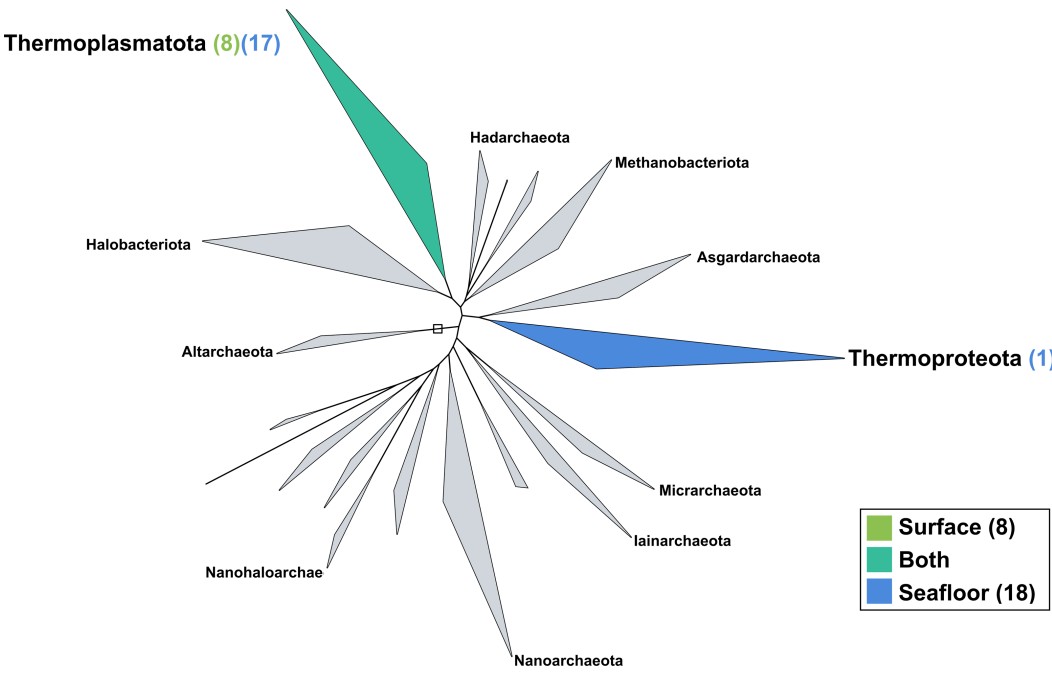

**Figure 5  Archaeal phylogenomic tree.** Distribution of the Marine Icelandic MAGs across 18 archaeal phyla from GTDB. The maximum likelihood tree was inferred from the concatenation of 122 proteins spanning a dereplicated set of 3,073 archaeal genomes (GTDB 05-RS95 released on the 17th July 2020) and the Marine Icelandic MAGs. Phyla containing MAGs from the surface seawater, seafloor or both are shown in green, blue or teal respectively. Number of Marine Icelandic MAGs from the surface and the seafloor in each phylum are indicated in between parenthesis in green and blue respectively.

the same genus, thus making it the most suitable binning algorithms out of the three tested in this study: CONCOCT, MaxBin2 and MetaBAT 2. This result is consistent with previous studies (*Yue et al., 2020*).

The ocean is a vast continuum and the samples were taken within a relatively small section/fraction of the North Atlantic Ocean at several sampling depths: the surface and the seafloor (90 m, 470 m, 1,006 m, and 1,060 m depending on the station). The differences in the sampling depth implies differences in lighting, pressure and temperature compared to the surface of the ocean. While the surface of the ocean is subjected to seasonal variations in day light and temperature, the seafloor remains darker and colder than the surface, and such parameters are driving microbial community structure and function. Therefore, we considered the surface and the seafloor of the ocean as two different types of environments which justifies our approach of two co-assemblies rather than assembling all of the 32 samples together. The fact that a number of MAGs were exclusively found in only one of the two environments, confirmed this.

## CONCLUSIONS

The goal of this study was to reconstruct MAGs from 31 samples from Icelandic sea waters. The 219 MAGs span across 13 bacterial and two archaeal phyla and contribute to

a more define picture of the global marine microbiome. Moreover, this study confirms, thanks to the inclusion of a mock community in the analysis, that the combination of co-assembly and binning with MetaBAT 2 allows, despite a relatively shallow sequencing depth, the recovery of quality MAGs that are a precious resource for further ecological and environmental studies.

## ACKNOWLEDGEMENTS

The authors would like to thank Kristinn Gudmundsson and Bjarni Saemundsson's crew from the Marine Research Institute, and Pauline Bergsten and Mia Cerfonteyn from the University of Iceland & Matís for sampling, Antonio Fernandez Guerra from the Max Plank Institute for Marine Microbiology and Arnar Pálsson from the University of Iceland for advice and Elvar Örn Jónsson from the University of Iceland for technical support. The analyses presented in the study were performed using the resources provided by the Icelandic High Performance Computing Centre at the University of Iceland.

### Funding

The work is part of the Microbes in the Icelandic Marine Environment (MIME) project which was funded by the Grant of Excellence (No. 163266-051) of the Icelandic Research Fund (Rannís). The funders had no role in study design, data collection and analysis, decision to publish, or preparation of the manuscript.

### Grant Disclosures

The following grant information was disclosed by the authors:
Grant of Excellence: 163266-051.
Icelandic Research Fund (Rannís).

### Competing Interests

Clara Jégousse, Pauline Vannier, René Groben and Viggó Marteinsson are employees of Matís ohf.

### Author Contributions

- Clara Jégousse conceived and designed the experiments, performed the experiments, analyzed the data, prepared figures and/or tables, authored or reviewed drafts of the paper, and approved the final draft.
- Pauline Vannier conceived and designed the experiments, performed the experiments, prepared figures and/or tables, authored or reviewed drafts of the paper, and approved the final draft.
- René Groben, Frank Oliver Glöckner and Viggó Marteinsson conceived and designed the experiments, authored or reviewed drafts of the paper, and approved the final draft.

## DNA Deposition

The following information was supplied regarding the deposition of DNA sequences:

Data are available at the ENA under project number PRJEB41565: all MAGs: ERS5621908 to ERS5622126; the surface and seafloor co-assemblies: ERS5565811 and ERS5565812.

## Data Availability

Code is available at Github:

https://github.com/clarajegousse/mime.

The following data are available at ENA:

- Raw data, co-assemblies and MAGs: PRJEB41565.

- Raw sequence data for the mock community: ERS5472810 to ERS5472840, and ERS5475418.

- The surface and seafloor co-assemblies: ERS5565811 and ERS5565812 respectively.

- MAGs: ERS5621908 to ERS5622126.

## Supplemental Information

Supplemental information for this article can be found online at http://dx.doi.org/10.7717/peerj.11112#supplemental-information.

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
