# Peer review of "A total of 219 metagenome-assembled genomes of microorganisms from Icelandic marine waters"

_PeerJ, doi:10.7717/peerj.11112_

## Round 0.1 · original submission · Minor Revisions

I believe that you will find the recommendations of the reviewers to be straightforward. While I do agree with the second reviewer that this would be a much more valuable contribution if it included some analysis and discussion, there is precedent for PeerJ to accept what is essentially a data report and thus it is left to you to decide to keep this in its current format or expand it.

·

Basic reporting

In their study, Jégousse and co-workers describe 219 archaeal and bacterial MAGs reconstructed from 31 metagenomes taken close to Iceland. The manuscript is well organized and written. It contains most of the information necessary for this type of article (see details below). The figures are appealing. The raw data has been deposited in a public repository.

Experimental design

Te manuscript is a data report. The methods section is fine but the information about the co assembly and the binning is missing (You need to check the figure and the results to llearn which tools were used). In addition, the authors should provide version numbers for all tools used (MegaHIT, CheckM, MetaBAT2, CONCOOT, MaxBin2, prodigal and so on). Also provide the version of the GTDB database.

Validity of the findings

All data has been provided with one exception. Please upload your MAGs to GenBank (the fasta file for each MAG). Otherwise this important information is not available to researchers.

Additional comments

Good Job. Only my previous comments and three minor comments:

Line 18: ... and sink lower depths of the ocean. -> Rephrase.
Line 23: (please change to) ... to bridge this knowledge gap.
Line 49: uncultured microorganisms (change to) -> yet uncultured marine lineages

Reviewer 2 ·

Basic reporting

All looks good.

Experimental design

The research question - describing a set of MAGs from Icelandic waters - is not relevant or meaningful. This is a nice dataset but there is no real research question in this manuscript, just a description of MAGs. It is too descriptive without posed or addressing an interesting research question. (See comments below.)

Validity of the findings

No comment

Additional comments

Jégousse and colleagues describe a unique metagenomic dataset, including 31 shotgun metagenomes from coastal Icelandic waters and a set of MAGs reconstructed from these samples. They included a mock marine microbial community in these analyses as well. The dataset described here is excellent and fills a needed hole in the databases of marine metagenomic data, but there is not much of a story in the manuscript other than describing the MAGs – it describes the dataset but does not emphasize question/conclusions other than “can we get MAGs from these samples.” This is perhaps best underscored in the first line of the Conclusions section: the goal of just “reconstruct[ing] MAGs from 31 samples from Icelandic sea waters” is not very exciting. I therefore believe there is little scientific merit in the manuscript as currently written. Yes, this is a great dataset (and a very nice MAG collection), but what does it mean? What can it teach us about marine microbiology? As currently written, I believe the manuscript is appropriate for a more descriptive journal (e.g., Microbiology Resource Announcements, Scientific Data) than a journal focused on research findings, but it has a TON of potential, as the metagenomes and MAGs are very nicely done and undoubtedly rich with information.

My suggestion is to use the manuscript as currently written as a starting point to write a longer, more detailed manuscript actually analyzing the MAGs. What are their read mapping patterns across samples? What interesting metabolic genes/pathways are present? Are there any connections between the distribution of MAGs and the unique physical setting of these samples? Etc. There could even be more of an emphasis on comparing the binning algorithms and recovery of the mock community, to make this more of a MAG methodology paper instead of a microbial ecology paper.

However, I will re-emphasize that this is a really nice set of metagenomes and MAGs, and I recognize the amount of hard work that the authors have undergone to get the dataset to this point. In my opinion this work would be put to much better use if the manuscript told an actual story or two instead of just describing the MAG set.

SPECIFIC COMMENTS
Lines 15-24: Give more detail of any results in the Abstract. It mostly just sets up why the samples are interesting, but I am more interested in what was found. Cut out some of the background on the study site, etc. to make more space if needed.

Lines 95-104: Add information here about quality-control steps (trimming low-quality sequences, etc.) as well as assembly and binning software and parameters. In particular, please make note of any software parameters that deviate from defaults. I realize some of this information is in Fig. 1B but it should also be in this section.

Lines 125-126: A comment – instead of just picking the MetaBat2 MAGs only, the authors could consider using DAS Tool (https://doi.org/10.1038/s41564-018-0171-1, https://github.com/cmks/DAS_Tool) to select a set of non-redundant MAGs from the results of all three binning algorithms. (Which may increase the diversity of MAGs in the final set.) It is nice to see more benchmarking of binning algorithms and mock community work.

Lines 116-182: No analysis of read mapping to MAGs across the metagenomes? Or genes/pathways present in the MAGs? I would really love to see this, in addition to knowing the phylogenetic identity of the MAGs. Without any of these types of analyses I don’t think this article has enough to justify publication as a research article (though as I note above, it could be published as a descriptive article just laying out the dataset).

Lines 174-176: This is confusing. MGIIa-L2 and -L1 are both within the family Ca. Poseidoniaceae. How many Ca. Poseidoniaceae MAGs were there?

Lines 190-192: CONCOCT and MaxBin2 also use both nucleotide composition and differential coverage to bin contigs, so this alone won’t explain why MetaBAT2 performs best on these samples…

---

## Round 0.2 · accepted · Accept

This is an interesting study, and valuable contribution.

---

## Author Rebuttal · Round 0.2

Matís ohf.

Vínlandsleið 12

112 Reykjavík, Iceland

3rd February 2021

Dear Editors,

We thank the reviewers for the valuable comments on the manuscript and have edited the manuscript to address their suggestions and comments.

In particular, we have uploaded all the data to the European Nucleotide Archive (ENA). Please find attached our responses to the reviewers´comments.

We hope the manuscript is now suitable for publication in PeerJ.

Clara Jégousse

Matís ohf. and the University of Iceland

On the behalf of the authors.

# Reviewer 1 (Bernd Wemheuer)

## Basic reporting

*In their study, Jégousse and co-workers describe 219 archaeal and bacterial MAGs reconstructed from 31 metagenomes taken close to Iceland. The manuscript is well organized and written. It contains most of the information necessary for this type of article (see details below). The figures are appealing. The raw data has been deposited in a public repository.*

We thank the reviewer for the positive feedback on the manuscript.

## Experimental design

*The manuscript is a data report. The methods section is fine but the information about the co assembly and the binning is missing (You need to check the figure and the results to llearn which tools were used). In addition, the authors should provide version numbers for all tools used (MegaHIT, CheckM, MetaBAT2, CONCOOT, MaxBin2, prodigal and so on). Also provide the version of the GTDB database.*

The subsection about Co-assembly and Binning was accidenty omitted in the manuscript version that was submitted and was added back into the manuscript (p. 3, lines 97-123). We also added the version numbers of the tools and the databases in the text and in Figure 1.

## Validity of the findings

*All data has been provided with one exception. Please upload your MAGs to GenBank (the fasta file for each MAG). Otherwise this important information is not available to researchers.*

We uploaded the MAGs (fasta files) as well as both co-assemblies (fastq files) in the ENA within project PRJNA693153. All MAGs are accessible using accession numbers ERS5621908-ERS5622126. The surface and seafloor co-assemblies are accessible using accession numbers ERS5565811 and ERS5565812 respectively.

## Comments for the Author

*Good Job. Only my previous comments and three minor comments:*

*Line 18: ... and sink lower depths of the ocean. -> Rephrase.*

We have rephrased the sentence (p. 1, lines 18-19): „The waters surrounding Iceland  are a complex environment where relatively warm salty waters from the Atlantic cool down and sink down to the deep.“

*Line 23: (please change to) ... to bridge this knowledge gap.*

We changed the sentence and added more information about the results in the introduction according to the suggestion of Reviewer 2 (p. 1, lines 21-26): „The resulting 219 MAGs include 191 bacterial, 26 archaeal and two eukaryotic MAGs to bridge the gap in our current knowledge of the global marine microbiome.“

*Line 49: uncultured microorganisms (change to) -> yet uncultured marine lineages*

We changed the sentence according to the suggestion (p. 2, line 51): „MAGs are particularly valuable for yet uncultured marine lineages“.

# Reviewer 2 (Anonymous)

## Basic reporting

*All looks good.*

## Experimental design

*The research question - describing a set of MAGs from Icelandic waters - is not relevant or meaningful. This is a nice dataset but there is no real research question in this manuscript, just a description of MAGs. It is too descriptive without posed or addressing an interesting research question. (See comments below.)*

Please see our response to this in the section „Comments for the Author" below.

## Validity of the findings

*No comment*

## Comments for the Author

*Jégousse and colleagues describe a unique metagenomic dataset, including 31 shotgun metagenomes from coastal Icelandic waters and a set of MAGs reconstructed from these samples. They included a mock marine microbial community in these analyses as well. The dataset described here is excellent and fills a needed hole in the databases of marine metagenomic data, but there is not much of a story in the manuscript other than describing the MAGs – it describes the dataset but does not emphasize question/conclusions other than "can we get MAGs from these samples." This is perhaps best underscored in the first line of the Conclusions section: the goal of just "reconstruct[ing] MAGs from 31 samples from Icelandic sea waters" is not very exciting. I therefore believe there is little scientific merit in the manuscript as currently written. Yes, this is a great dataset (and a very nice MAG collection), but what does it mean? What can it teach us about marine microbiology? As currently written, I believe the manuscript is appropriate for a more descriptive journal (e.g., Microbiology Resource Announcements, Scientific Data) than a journal focused on research findings, but it has a TON of potential, as the metagenomes and MAGs are very nicely done and undoubtedly rich with information.*

*My suggestion is to use the manuscript as currently written as a starting point to write a longer, more detailed manuscript actually analyzing the MAGs. What are their read mapping patterns across samples? What interesting metabolic genes/pathways are present? Are there any connections between the distribution of MAGs and the unique physical setting of these samples? Etc. There could even be more of an emphasis on comparing the binning algorithms and recovery of the mock community, to make this more of a MAG methodology paper instead of a microbial ecology paper.*

*However, I will re-emphasize that this is a really nice set of metagenomes and MAGs, and I recognize the amount of hard work that the authors have undergone to get the dataset to this point. In my opinion this work would be put to much better use if the manuscript told an actual story or two instead of just describing the MAG set.*

We agree with the reviewer that a paper with an in-depth focus on ecological patterns, metabolic pathways, etc. based on our dataset would be a very interesting and valuable contribution to the knowledge about Arctic marine microbiology.

However, we decided to publish this work in its current form as the description of the sampling, data collection and reconstruction of the MAGs. We believe that such straight forward description makes it easier for other scientitst to (re)use this dataset as a resource. We considered PeerJ as a suitable journal to publish this descriptive work as we had read

similar work published in PeerJ in the past. Further analyses, including those that will address the reviewers suggestions and comments, are currently been conducted and will be submitted for publication in a follow-up paper in the near future.

*SPECIFIC COMMENTS*

*Lines 15-24: Give more detail of any results in the Abstract. It mostly just sets up why the samples are interesting, but I am more interested in what was found. Cut out some of the background on the study site, etc. to make more space if needed.*

We added a sentence summarising the results in the introduction (p.1, lines 25-26): „The resulting 219 MAGs include 191 bacterial, 26 archaeal and two eukaryotic MAGs to bridge the gap in our current knowledge of the global marine microbiome."

*Lines 95-104: Add information here about quality-control steps (trimming low-quality sequences, etc.) as well as assembly and binning software and parameters. In particular, please make note of any software parameters that deviate from defaults. I realize some of this information is in Fig. 1B but it should also be in this section.*

We added the version numbers for all softwares and specified parameters throughout the text and in Figure 1.

*Lines 125-126: A comment – instead of just picking the MetaBat2 MAGs only, the authors could consider using DAS Tool (https://doi.org/10.1038/s41564-018-0171-1, https://github.com/cmks/DAS_Tool) to select a set of non-redundant MAGs from the results of all three binning algorithms. (Which may increase the diversity of MAGs in the final set.) It is nice to see more benchmarking of binning algorithms and mock community work.*

We thank the reviewer for pointing out DAS Tool and we tested it. However, results from DAS Tool did not include the *Photobacterium* of the mock community while MetaBAT2 did for both co-assemblies and we therefore decided to use the results of MetaBAT2 and not include the additional results from DAS Tool into the manuscript.

*Lines 116-182: No analysis of read mapping to MAGs across the metagenomes? Or genes/pathways present in the MAGs? I would really love to see this, in addition to knowing the phylogenetic identity of the MAGs. Without any of these types of analyses I don't think this article has enough to justify publication as a research article (though as I note above, it could be published as a descriptive article just laying out the dataset).*

As we wrote in our reply in the Comments section, we deliberately decided to outline this publication as a descriptive paper. The analyses and data that the reviewer mentions will be addressed in a follow-up paper in the near future.

*Lines 174-176: This is confusing. MGIIa-L2 and -L1 are both within the family Ca. Poseidoniaceae. How many Ca. Poseidoniaceae MAGs were there?*

We agree with the reviewer that the sentence could have been misunderstood and we rephrased it to be clearer (p. 5, lines 204-207): „The surface co-assembly yielded only eight archaeal MAGs, all within the Thermoplasmota phylum, including three MAGs within the genus MGIIb-O2 of the Thalassarchaeaceae family and five within the Poseidoniaceae family."

*Lines 190-192: CONCOCT and MaxBin2 also use both nucleotide composition and differential coverage to bin contigs, so this alone won't explain why MetaBAT2 performs best on these samples…*

We agree with the reviewer that our explanation was too brief and therefore did not address the distinction between the two algorithms adequately. As the comparison between binning tools was adressed in detail in the paper by Yue et al. (which we cite) and is not the focus of our paper, we decided to delete this part from the manuscript.